# Intensity Ratio of *Kβ*/*Kα* in Selected Elements from Mg to Cu, and the Chemical Effects of Cr *Kα*_1,2_ Diagram Lines and Cr *Kβ*/*Kα* Intensity Ratio in Cr Compounds

**DOI:** 10.3390/ijms24065570

**Published:** 2023-03-14

**Authors:** Yoshiaki Ito, Tatsunori Tochio, Michiru Yamashita, Sei Fukushima, Takashi Shoji, Katarzyna Słabkowska, Łukasz Syrocki, Marek Polasik, Jana Padežnik Gomilsek, José Pires Marques, Jorge Miguel Sampaio, Mauro Guerra, Jorge Machado, José Paulo Santos, Assala Hamidani, Abdelhalim Kahoul, Paul Indelicato, Fernando Parente

**Affiliations:** 1Laboratory of Atomic and Molecular Physics, ICR, Kyoto University, Gokasho, Uji 611-0011, Japan; 21-24-14 Inadera, Amagasaki 661-0981, Japan; 3Hyogo Prefectural Institute of Technology (HIT), 3-1-12 Yukihira, Suma-ku, Kobe 654-0037, Japan; 4Kobe Material Testing Laboratory Co., Ltd., 47-13 Nijima, Harima-cho, Kako-gun 675-0155, Japan; 5Rigaku Corporation, 14-8 Akaoji-cho, Takatsuki 569-1146, Japan; 6Faculty of Chemistry, Nicolaus Copernicus University in Toruń, Gagarina 7, 87-100 Toruń, Poland; 7Institute of Plasma Physics and Laser Microfusion, Hery 23, 01-497 Warsaw, Poland; 8Faculty of Mechanical Engineering, University of Maribor, Smetanova 17, SI-2000 Maribor, Slovenia; 9LIP—Laboratório de Instrumentação e Física Experimental de Partículas, Av. Prof. Gama Pinto 2, 1649-003 Lisboa, Portugal; 10Faculdade de Ciências, Universidade de Lisboa, 1749-016 Lisboa, Portugal; 11Laboratório de Instrumentação, Engenharia Biomédica e Física da Radiação (LIBPhys-UNL), Departamento de Física, Faculdade de Ciências e Tecnologia da Universidade Nova de Lisboa, Monte da Caparica, 2892-516 Caparica, Portugal; 12Department of Matter Sciences, Faculty of Sciences and Technology, Mohamed El Bachir El Ibrahimi University, Bordj Bou Arreridj 34030, Algeria; assala.hamidani@univ-bba.dz (A.H.);; 13Laboratory of Materials Physics, Radiation and Nanostructures (LPMRN), Mohamed El Bachir El Ibrahimi University, Bordj Bou Arreridj 34030, Algeria; 14Laboratoire Kastler Brossel, Sorbonne Université, CNRS, ENS-PSL Research University, Collège de France, Case 74, 4 Place Jussieu, 75005 Paris, France

**Keywords:** atomic fundamental parameters, X-ray intensity ratios, chemical effects on intensity ratios

## Abstract

Kα,β X-ray lines from photon excitation were measured in selected elements from Mg to Cu using a high-resolution double-crystal X-ray spectrometer with a proportional counter, and the Kβ/Kα intensity ratio for each element was obtained, after correcting for self-absorption, detection efficiency, and crystal reflectance. This intensity ratio increases rapidly from Mg to Ca but, in the 3d elements region, the increase becomes slower. This is related to the intensity of the Kβ line involving valence electrons. The slow increase of this ratio in the 3d elements region is thought to be due to the correlation between 3d and 4s electrons. Moreover, the chemical shifts, FWHM, asymmetry indices, and Kβ/Kα intensity ratios of the Cr compounds, due to different valences, were also investigated using the same double-crystal X-ray spectrometer. The chemical effects were clearly observed, and the Kβ/Kα intensity ratio was found to be compound-dependent for Cr.

## 1. Introduction

Since the early days of X-ray spectroscopy, the Kβ/Kα X-ray intensity ratio for elements across the periodic table has been extensively studied experimentally, without taking into account the resolution in the X-ray spectrometer. Due to advances in solid-state X-ray detectors, such as Si(Li) or intrinsic Ge detectors, much experimental data have been reported for *K* X-ray emission following radioactive decays, photon irradiation, and charged-particle impact. Recently, many experimental data were compiled, and recommended average values were proposed by Hamidani et al. [1]. The Kβ/Kα X-ray intensity ratio is frequently considered to be a characteristic quantity for each element, except in the cases where heavy-ion bombardment is used, leading to multiple ionizations [2]. From the comparison of the measured values with the theoretical results for free atoms, good agreement was found with the relativistic calculations of Scofield [3].

Paic and Pecar [4] found that, for first-row transition elements, the Kβ/Kα intensity ratio depends on the mode of excitation. The difference between the ratios for electron-capture decay and for photoionization is of almost 10%. Similar results were obtained by Arndt et al. [5], who pointed out that this difference comes from a strong shake-off process accompanying photoionization. When the ionized electron comes from a core shell or subshell, there is a probability that one or more additional electrons may be excited or ionized, leading to multiple-hole configurations. The contribution of the [1s3d] shake processes, where $\left[nln^{\prime }l^{\prime }\right]$ henceforth denotes the orbitals where the holes were created, to the full-width at half-maximum (FWHM), measured using photon excitation, was recently reported by Ito et al. [6,7], and Deutsch et al. [8], in the Kα and Kβ line profiles of 3d elements. Using ionization by electron bombardment, Hölzer et al. [9] found that the contribution of this shake process was present throughout the diagram lines. On the other hand, several experiments have been performed to investigate the chemical effect on the Kβ/Kα intensity ratios for 3d elements [10,11,12]. Brunner et al. [13], who used protons as the ionization method, explained their experimental results by a contraction of 3p orbitals due to the charge delocalization of the 3d valence electrons. They also pointed out that the chemical effect is almost of the same order of magnitude as the effect of the excitation mode, and both effects should be studied separately.

Urch [14] discussed the chemical effect on the *K* X-ray spectra, based on molecular-orbital (MO) theory. Similar studies were already extensively made by Meisel et al. [15], and Barinki and Nefedow [16]. The latter studies were mostly concerned only with transition energies, diagram lines FWHMs and asymmetric profiles, and no quantitative discussions on the intensities were had. Mukoyama et al. [17,18] tried to determine the Kβ/Kα intensity ratio in compounds, but the monochromator used was a θ−3θ system, distorting the X-ray profile [19] and, consequently, making difficult to obtain this intensity ratio accurately, since the $K\alpha_{2}/K\alpha_{1}$ diagram lines intensity ratio was obtained based on asymmetric fitting analysis.

Unlike the θ−3θ method in the experiment described above, a high-resolution double-crystal X-ray spectrometer with the θ−4θ method was used in this work, allowing for the instrumental function to be evaluated correctly, to measure without distortion the diagram line. In addition, since a proportional counter (PC) is used, the latter method accurately determines the detection efficiency for each energy. In this study, we first measured together the Kα and Kβ diagram lines of the elements with atomic numbers from Z=12 to 29, and analyzed them using the asymmetric Lorentzian fitting method, in order to obtain the Kβ/Kα intensity ratio. The results were compared with theoretical calculations performed by us using two different multi-configuration Dirac-Fock (MCDF) methods, in order to assess the quality of the calculated data. Additionally, we measured the Kβ and Kα X-ray intensity for Cr compounds, using the same double-crystal spectrometer, and evaluated the chemical effects on the Kβ/Kα ratios.

## 2. Results and Discussion

### 2.1. The Kβ/Kα Intensity Ratio in the Elements from Mg to Cu

In contrast to the $K\alpha_{2}/K\alpha_{1}$ intensity ratio, almost all published measurements of the $K\beta_{1,3}/K\alpha_{1,2}$ intensity ratio were performed using semiconductor detectors (SD), since the energy difference between the Kβ and Kα complexes, 500–900 eV for the transition elements, is considerably larger than the 200 eV energy resolution of the SD. We were able to find in the literature only a single experiment using a high-resolution crystal spectrometer (CS) [9] with a SD.

The double-crystal spectrometer (DCS) with the proportional counter is much more suitable than CS for excitation by photons, due to its much higher resolution, and the almost absence of background due to bremstrahlung, allowing for a clear separation of X-ray satellite lines, as seen in Figure 1. As can be seen from that figure, the background due to the photon excitation is almost negligible.

The intensity ratios and asymmetries of the $K\alpha_{1,2}$ lines for each element, obtained in this work, are shown in Figure 2a,b, respectively, together with previously reported data. The asymmetry index is defined as the ratio of the width of the low-energy part to the width of the high-energy part of the half-width [20,21]. Generally, the asymmetry index is larger when using higher order Bragg reflections. If we use Si(220) and Si(440) crystals in the X-ray spectrometer, the asymmetry index should be larger in the latter, as the influence of the profile base is smaller. The DCS measured $K\alpha_{1,2,3,4}$ (the $K\alpha_{3,4}$ lines resulting from KL double ionization) and $K\beta_{1,3}$ spectra were recorded in only a single run for each element, with the exception of Mg, Al, and Si $K\alpha_{1,2,3,4,5,6}$ (the $K\alpha_{5,6}$ lines resulting from KLL triple ionization) and Kβ spectra. As can be seen from Figure 2a, in the case of 3d elements the $K\alpha_{2}/K\alpha_{1}$ intensity ratio fluctuates around the purely statistical 0.5 value due to the contribution of the [1s3p] or [1s3d] shake processes. Although Hölzer et al. [9] obtained the $K\alpha_{1}$ and $K\alpha_{2}$ intensity ratio for each element, they are not referenced in Figure 2a since their intensity ratios were obtained by a different fitting method. Chantler et al. [22] also increased the number of peaks in order to obtain zero residuals in the multiple peaks fitting of the $K\alpha_{1,2}$ lines and did not obtain an intensity ratio taking into account the contribution of the shake process.

Using an asymmetry fitting analysis, the area intensities of $K\alpha_{1}$, $K\alpha_{2}$, $K\alpha^{\prime \prime }$ (satellites resulting from the [1s3p] shake processes), $K\alpha_{3,4}$, and $K\beta_{1,3}$ were determined to obtain the $K\beta_{1,3}/\left(K\alpha_{1,2,3,4}+K\alpha^{\prime \prime }\right)$ or $K\beta_{1,3}/K\alpha_{1,2,3,4}$ intensity ratios, as shown in Figure 1. Hölzer et al. [9] obtained $K\beta_{1,3}/K\alpha_{1,2}$ intensity ratios, from Cr to Cu, that may be compared with our $K\beta_{1,3}/K\alpha_{1,2,3,4}$ intensity ratios. From Fe to Cu the difference between the theoretical ratios and the corrected measured intensity ratios increases. This may result from the fact that the experimental data include satellite intensities unlike the theoretical calculations. We found that the inclusion of all satellites in the $K\beta_{1,3}/K\alpha_{1,2}$ intensity ratios calculations increases the results by 69.2% for Al, 3.79% for Ca, 3.01% for Ti, and 3.65% for Cu, for instance.

Figure 2a shows a large spread in the $K\alpha_{2}/K\alpha_{1}$ intensity ratios, about 0.53 for Sc to V, and about 0.48 for Mn and Fe. The values of this ratio for elements Sc to V can be attributed to the contribution of the [1s3p] shake process, which leads to the $K\alpha^{\prime \prime }$ satellite, that can be seen, for example, in the Ti Kα,β line of Figure 1, because this shake process contributes to the higher energy side intensity of each diagram line. In other words, the asymmetric indices of the $K\alpha_{1,2}$ spectra is less than 1.0 because there is no contribution from the [1s3d] shake process in Ca and very little [1s3p] contribution in Cr. For elements above Cr, the $K\alpha_{1,2}$ spectral lines are influenced by the contribution of the [1s3d] shake process, so that the $K\alpha_{2}/K\alpha_{1}$ intensity ratio for Mn and Fe is about 0.48. The contribution of the [1s3d] shake process to the lines in the figure is on the low energy side. Compared to the variation of the $K\alpha_{2}/K\alpha_{1}$ intensity ratio of the 3d elements, those of Mg, Al, and Si are relatively small, and the hidden satellites due to the shake process do not affect this ratio so much. In contrast, those of 3d elements are affected by the hidden satellites.

The formula used for the corrected ratio (corr) as a function of the measured ratio (meas) is
(1)$$\frac{I_{K\beta_{\mathrm{corr}}}}{I_{K\alpha_{\mathrm{corr}}}}=\frac{I_{K\beta_{\mathrm{meas}}}}{I_{K\alpha_{\mathrm{meas}}}}\times \frac{S_{\alpha }}{S_{\beta }}\times \frac{D_{\alpha }}{D_{\beta }}\times {\left(\frac{R_{\alpha }}{R_{\beta }}\right)}^{2},$$
where *S*, *D*, and *R* are the self-absorption, the detector efficiency, and the crystal integrated reflectivity, respectively. The corrected values for the Kβ/Kα intensity ratios were calculated according to Equation (Equation 1), and are given in Table 1. For example, in the case of Cu the measured intensity ratio is 0.1617, the detection efficiency for the $K\alpha_{1}$ line is 0.94, and that for the $K\beta_{1,3}$ lines is 0.904. The detection efficiency ratio D(Kβ/Kα) is therefore 1.04. The self-absorption ratio S(Kβ/Kα)0.756 for each of those lines, and the reflection intensity ratio $R^{2}\left(K\beta /K\alpha \right)\;1.17$ for two reflections. The correction value is therefore 0.1487. The correction values for the other elements were obtained using this procedure.

The Kβ and Kα lines were measured simultaneously. Furthermore, the Cu $K\alpha_{1,2}$ line intensities were repeatedly measured for each Kα,β line measurement, in order to check the intensity variation in the spectrometer. This allowed for a better assessment of the intensity. The uncertainty in the peak positions of Cu $K\alpha_{1}$ and $K\alpha_{2}$ emission lines was estimated as less than 0.1%. The energy values of the Kα lines determined by the multiple peak analysis are in good agreement with those of Bearden [27], and Deslattes et al. [28] (Table 2).

The ability to accurately measure X-ray profiles is one of the features of this wide-area scanning double-crystal X-ray spectrometer. For Cu measurements, the uncertainty of the $K\alpha_{2}/K\alpha_{1}$ intensity ratio is about 0.6%, while the uncertainty of the $K\beta_{1,3}/K\alpha_{1,2}$ intensity ratio is about 1.6%. The normalized intensity of the Kα and Kβ lines were calculated by fitting asymmetric Lorentz functions to each element, except for Mg, Al, and Si, where symmetric Lorentz functions were used due to the lack of asymmetry in the $K\alpha_{1,2}$ diagram lines of these elements.

Table 1 shows the corrected Kβ/Kα intensity ratios for each element in this study, together with the values of Salem et al. [25], Ertugrul et al. [26], Hölzer et al. [9], the recommended values of Hamidani et al. [1], obtained from the average over a large number of previous experimental values, and the theoretical values obtained in this work, and by Scofield [3]. Our systematic MCDF (GRASP) calculations were performed for various valence electronic configurations ($3d^{n-2}4s^{2}$ and $3d^{n-1}4s^{1}$, where *n* is the number of valence electrons) for 3d elements. On the other hand, the MCDFGME calculations were performed including only the diagram line intensities and, in selected cases, the diagram and satellite intensities, as shown, in the 11th and 12th columns.

Kβ lines of all 3rd period elements result from electron transitions from an occupied state in a valence orbital (valence band) to an inner shell (1s orbital). In the free neutral Mg atom, only 3s orbitals (valence electrons) are occupied. However, for metallic Mg, and due to the low energy difference between the 3s and 3p levels, the latter are also contained in the valence band (occupied band). The hybrid nature of these orbitals allows that electron transitions between 3p and 1s levels are possible in metallic Mg, leading to the presence of Kβ lines in the measured spectra. This possibility was not taken in account in our theoretical calculations.

In Figure 3, our Kβ/Kα intensity ratio values are compared with previous theoretical results and experiments employing electron or photon excitation only, since the pronounced multiple ionization that occurs in the case of heavy-ion excitation strongly modifies this ratio [1,4]. In the same figure, the recommended values of Hamidani et al. [1] are also plotted.

As seen in this figure, the intensity ratio increases comparatively abruptly from Mg to Si and then to Ca, and the trend becomes slower for 3d elements. The reason for this may be related to the exchange interaction between 2p and 3d electrons, and between 3p and 3d electrons. The calculated Kβ/Kα intensity ratios are in good agreement with the recommended experimental and theoretical values. The intensity ratios, measured in this study, of the 3d elements, except for Ca, are exponentially fitted to the solid lines shown in Figure 3. For atomic numbers above Fe the experimental values of the present study agree well with those of Hölzer et al. [9] obtained with a high-resolution single-crystal spectrometer, although they also show differences from previous experimental and calculated values, as well as from the calculated values obtained in the present study. Moreover, the two solid curves in Figure 3 are linear fittings of the measured Kβ/Kα intensity ratios from Mg to Si, and from Sc until Cu.

The slope of the Kβ/Kα intensity ratio increase changes around the onset of 3d elements. It is noteworthy that for these elements, the ratio agrees better with the value of the $3d^{n-2}4s^{2}$ coordination interaction marked by open circles in Figure 3, calculated in this study, than with the recommended average value. This value is slightly different from the one obtained by Scofield, and is the first calculation that shows the need to take into account the configuration interaction between 3d and 4s electrons.

Our method is presented as a useful method for safety and security: trivalent Cr is relatively safe, but hexavalent Cr can cause dermatitis and tumors if left adhering to the skin and mucous membranes. Drinking contaminated well water causes vomiting. In addition, we are surrounded by Cr-plated metals, such as Cr-plating. Thus, in order to identify the amount of hexavalent Cr present, a method to identify trivalent and hexavalent Cr is an urgent issue.

The aforementioned asymmetry index, chemical shifts, and FWHMs are used to evaluate the chemical effects of Cr compounds, as follows.

### 2.2. Cr Compounds

The measurement conditions are shown in Table 3. The X-ray spectra were measured with a high-resolution double-crystal spectrometer. In this spectrometer, unlike the case of a single crystal spectrometer, the horizontal broadening is suppressed by the second crystal, so it is sufficient to use only the slit to suppress the vertical broadening. The spectrometer used in this study is equipped with a Soller slit with a slit length of 100 mm and a spacing of 1 mm between each layer. The detector was a gas flow proportional counter, and PR10 gas Ar${}_{0.9}$(CH${}_{4}$)${}_{0.1}$ was used. The measurement time per point for $K\alpha_{1}$ and $K\alpha_{2}$ spectra, without $K\alpha_{3,4}$ satellites was 5 s for the metal and Oh-symmetric compounds, and 6 s for the Td-symmetric compounds, due to their rather weak intensity (as seen in Figure 4). The 2θ step angle of this spectrometer is 0.0005°. The measurement time per point for $K\beta_{1,3,5}$ spectra are 90 s for Cr metal and Cr${}_{2}$O${}_{3}$, 150 s for K${}_{2}$Cr${}_{2}$O${}_{7}$, FeCr${}_{2}$O${}_{4}$, and CoCr${}_{2}$O${}_{4}$, and 260 s for K${}_{2}$CrO${}_{4}$. The 2θ step angle of this spectrometer is 0.002°. Since the Kβ spectral lines of the Cr compounds are extremely weak, the Kα,β lines could be measured separately.

The results of fitting in each measured spectrum with two asymmetric Lorentz functions are shown in Table 4. As mentioned above, spectra measured with a double-crystal X-ray spectrometer are less affected by the instrument function. The uncorrected values for the broadening effect of the instrumental function are given in Table 4. The reason for not correcting is that, for example, the $K\alpha_{1}$ peak of an Oh-symmetric compound may be likely to contain some peaks, each of which is affected as shown in Figure 4, and it is difficult to distinguish the peaks in the spectral profile. Therefore, we used the values obtained by analyzing the observed data with asymmetric fitting, without any corrections for the instrument function. The present measurement results are in close agreement with previous ones [29,30,31,32], such as the asymmetry and large half-width of the $K\alpha_{1}$ peak in Oh-symmetric compounds [29,32], and the asymmetry and large half-width of the $K\alpha_{2}$ peak in Td-symmetric compounds [29,31]. In comparison with the results of [29] (Cr metal, K${}_{2}$CrO${}_{4}$, Cr${}_{2}$O${}_{3}$ only) (Table 5) on the energy shifts of $K\alpha_{1}$ and $K\alpha_{2}$ peaks with respect to the metal, they are qualitatively consistent in many aspects, such as the energy shifts of $K\alpha_{1}$ and $K\alpha_{2}$, and the energy difference between $K\alpha_{1}$ and $K\alpha_{2}$ peaks.

In addition, the compound effect for Cr is clearly seen in the Kβ/Kα intensity ratio (see Table 4). In other words, the compound with Td symmetry and the compound with Oh symmetry can be distinguished. Moreover, the changes in the values of the Kβ/Kα intensity ratio can be associated not only with symmetry but also with the valence of Cr in these compounds. In the K${}_{2}$CrO${}_{4}$ and K${}_{2}$Cr${}_{2}$O${}_{7}$ compounds, Cr has a valence equal to +6, but for the remaining compounds (in Table 4), Cr has a valence of +3. It turns out that the Kβ/Kα intensity ratio calculated by us (using GRASP package) for Cr with the valence of +6 (3d${}^{0}$4s${}^{0}$ configuration) is 0.1610, and for Cr with the valence of +3 (3d${}^{3}$4s${}^{0}$ configuration) is 0.1395. As can be seen, the obtained theoretical values are in good agreement with the experimental results presented in Table 4. Similarly, for metallic Cr, our calculated Kβ/Kα intensity ratios are 0.1333 and 0.132, as seen in Table 1 in agreement with the experimental value of 0.132.

## 3. Methods and Materials

### 3.1. Experimental Procedure

The fluorescence Kα,β X-ray spectra of some elements from Mg to Cu were measured using a RIGAKU double-crystal spectrometer (System 3580E, Rigaku Corporation, Takatsuki, Japan) and the photon excitation method. The instrumental function is explained in detail in refs. [6,7,20,24]. An end-window type was adopted as the primary X-ray source without the contamination of the filament material, usually rhodium. The spectra were excited by primary X-rays from a Rh-target X-ray tube operated at 40 kV and 60 mA. Experimental conditions are listed in Table 6. Some of the observed Kα,β emission spectra for elements Mg to Cu are shown in Figure 1. The symmetric Si(220) and ADP(101) reflections were used in both crystals. As targets, we used CaF${}_{2}$ crystal powder for Ca, foils for Sc, V, Co, and Ni, plates for Mg, Al, Ti, Mn, Fe, and Cu, and wafer for Si. The Cr compounds Cr${}_{2}$O${}_{3}$, K${}_{2}$CrO${}_{4}$, and K${}_{2}$Cr${}_{2}$O${}_{7}$ were reagents (Nacalai), each with special grade purity, while FeCr${}_{2}$O${}_{4}$ and CoCr${}_{2}$O${}_{4}$ were synthesized.

Compounds with two spinel-type structures, FeCr${}_{2}$O${}_{4}$ and CoCr${}_{2}$O${}_{4}$, were synthesized using the following method:Cr${}_{2}$O${}_{3}$ and FeO(CoO) are mixed in an agate mortar and ground well.Vacuum-seal the mixture in a glass tube and sinter it.Measure of the X-ray diffraction pattern of the resulting sample and repeat steps 1 and 2 until there are no extra peaks.

Actually, we repeated steps 1 and 2 twice, for both spinels. The sample was placed on a greased aluminum plate. When the aluminum plate was set in the sample holder, the surface was covered with a 6 μm thick Mylar film, to prevent the sample from scattering in the chamber. The crystal structures were confirmed by X-ray diffraction. Since the intensity of Cr oxide is weak, W target was used as the primary X-ray source 40 kV with tube voltage and 70 mA tube current.

The spectra were measured using an Ar${}_{0.9}$(CH${}_{4}$)${}_{0.1}$ gas flow for the elements Mg to Si, and a sealed Xe gas proportional counter with a dead time of less than 1 μs, for the other elements. The window thickness of the PC detectors we used was precisely guaranteed, and in the case of the Sealed PCs, we used detectors whose internal pressure was guaranteed by the manufacturer. The intensity of fluorescent X-rays due to primary excitation is less than 5000 counts/s, so in the detector does not occur counting loss. In order to evaluate the stability of the X-ray spectrometer as well, we performed the measurements of the Ca to Cu Kα,β lines in the following order of measurement: Cu $K\alpha_{1,2}$ lines–Ca Kα,β lines–Cu $K\alpha_{1,2}$ lines–⋯–Cu $K\alpha_{1,2}$ lines–Cu Kα,β lines–Cu $K\alpha_{1,2}$ lines.

In this study, the $K\alpha_{1,2}$ and $K\beta_{1,3}$ lines of the 3d elements were fitted with asymmetric Lorentz functions. For the elements below Ca, the $K\alpha_{1,2}$ and Kβ lines were fitted with symmetric Lorentz functions, as the effect of the shake process does not appear in the profile. In the fitting process, four parameters were used for each asymmetric peak: energy, half-width, asymmetry, and relative intensity. For the symmetric peaks, the parameters are energy value, half-width, and intensity. However, Al and Si Kb lines are difficult to fit even with two symmetric Lorentz function according to the density of the states of 3p electrons in Al. This is an issue to address in the future.

Table 2 shows the results of fitting the Cu $K\alpha_{1,2}$ lines with two asymmetric and four symmetric Lorentz functions, which were evaluated to check the stability of the instrument. In this table, $K\alpha_{1}-K\alpha_{2}$ is the energy difference between the peak values of the $K\alpha_{1}$ and the $K\alpha_{2}$ lines; the four fittings are the fitting of the $K\alpha_{1}$ and $K\alpha_{2}$ lines with two Lorentz functions each; $K\alpha_{12}$ and $K\alpha_{22}$ are satellite lines arising from the [1s3d] shake process and their intensity is weak. $K\alpha_{11}$ and $K\alpha_{21}$ are the characteristic (diagram) lines. The contribution of the instrumental function was taken in account for these lines.

As can be seen from Figure 5, the device was very stable during the spectra measurements. This spectrometer has a high resolution and the ability to scan a range of 20° to 147° in 2θ angle. Temperature in the X-ray spectrometer chamber was controlled within $35.0\pm 0.5\;{}^{{}^{\circ}}C$, and the vacuum system under ∼5 Pa. Neither smoothing nor correction were applied to the raw data. The spectrometer vertical divergence slit is 0.573°. X-rays from the primary source are irradiated onto the sample, the X-rays from the sample pass through a Soller slit to prevent vertical divergence and are then spectrally split by the first crystal. The two crystals are linked so that they form a constant angle (π−2θ) to prevent the spectra from changing shape. For the energy calibration, the values of Bearden [27] were used, as references, for Mg, Al, and Si diagram lines, and those of Deslattes et al. [28] for the diagram lines of the other elements. The Kβ/Kα intensity ratio was corrected for self-absorption, crystal reflectivity, and detection efficiency. Mass absorption coefficients of elemental and compound materials were taken from the XCOM database (NIST) [33].

### 3.2. Theoretical Procedure

The systematic theoretical study realized in this paper was performed using the multiconfigurational Dirac-Fock (MCDF) method in two approaches: the General-purpose Relativistic Atomic Structure Package (GRASP), and the MCDFGME code. The GRASP approach was mainly developed by Grant and co-workers and is described in detail in many papers [34,35,36,37,38,39,40,41,42,43,44,45,46,47,48,49]. Moreover, all basic ideas of the alternative Special Average Level (SAL) version of MCDF calculations, which is used in this work, were presented by Polasik [48]. The MCDFGME code was developed by Desclaux and Indelicato [50,51] and have had many improvements since its creation, such as the inclusion oƒ QED corrections, namely the Uelhing potential, in the self-consistent field allowing for its evaluation to all orders.

In the MCDF method, the effective relativistic Hamiltonian for the *N*-electron atom is taken in the form (atomic units are used)
(2)$$\hat{H}=\sum_{i=1}^{N}{\hat{h}}_{D}\left(i\right)+\sum_{j>i=1}^{N}V_{B}\left(i,j\right)$$
where ${\hat{h}}_{D}\left(i\right)$ is the one-electron Hamiltonian,
(3)$${\hat{h}}_{D}\left(i\right)=\alpha_{i}\cdot p_{i}+\left(\beta_{i}-1\right)+V_{i}.$$

Here, $\alpha_{i}$ and $\beta_{i}$ are 4×4 Dirac matrixes, and $V_{i}$ describes the interaction of one electron with the atomic nucleus. The term $V_{B}\left(i,j\right)$ describes the interaction between the *i*-th and the *j*-th electron, i.e., a sum of Coulomb interaction operator and the Breit operator (due to transversely polarized photons).

The wave function for a state of a *N*-electron atom (characterized by the quantum numbers determining the value of the square of the total angular momentum *J*, projection of the angular momentum on the chosen direction *M*, and parity *p*) is assumed, in the MCDF method, to be the linear combination
(4)$$\Psi_{s}\left(JM^{p}\right)=\sum_{m}c_{m}\left(s\right)\Phi \left(\gamma_{m}JM^{p}\right),$$
where $\Phi \left(\gamma_{m}JM^{p}\right)$ are *N*-electron configuration state functions (CSF), $c_{m}\left(s\right)$ are the configuration mixing coefficients for state *s*, and $\gamma_{m}$ represents all information required to uniquely define a certain CSF. The function $\Phi \left(\gamma_{m}JM^{p}\right)$ is a *N*-electron function given in the form of Slater determinant or combination of Slater determinants built from one-electron Dirac spinors.

Moreover, in order to obtain very high accuracy, in the MCDF calculations, apart from the transverse Breit interaction, it is necessary to also consider quantum electrodynamic (QED) corrections to the energy, i.e., self-energy, and vacuum polarization. Using a finite-size nucleus model during the calculations was also crucial, including a two-parameter Fermi charge distribution.

In the MCDFGME approach, full relaxation was used, i.e., the wave functions for the transitions initial and final states were calculated independently in monoconfiguration mode, the corresponding non-orthogonality effects being treated by the formalism proposed by Löwdin [52]. The inclusion of relaxation was found to be of utmost importance for the reliability of the results [53]. The length gauge was used for all radiative transition probabilities. In the present calculation, the Breit interaction and the vacuum polarization terms were included in the self-consistent field process, while retardation and QED effects, namely the self-energy, were treated as perturbation.

## 4. Conclusions

A high-resolution double-crystal X-ray spectrometer with a proportional counter is used to measure Mg to Cu Kα,β diagram lines by photon excitation, and the Kβ/Kα intensity ratios for these elements were obtained by correcting for self-absorption, detection efficiency, and crystal reflectance. From Mg to Ca, this intensity ratio increases rapidly with an increasing number of 3p electrons, while in the region from Sc to Cu the increase is very slow. The intensity ratios measured with the high-resolution crystal spectrometers show a slow and increasing trend above Fe, but the present calculations, which take into account the interaction between 3d and 4s, also show significant differences; probably because, in the latter case, satellite intensities are not included. This is a subject for future research. We also investigated the chemical shifts, FWHM, asymmetry indexes, and Kβ/Kα intensity ratios for Cr compounds of different valence, using the same double-crystal X-ray spectrometer. Chemical effects were clearly observed for these different valences, and the Kβ/Kα intensity ratios were Cr compound dependent. Furthermore, the experimental intensity ratios could be deduced from theoretical calculations of the intensity ratios for the Cr 3+ and 6+ electronic states only.

## Figures and Tables

**Figure 1 ijms-24-05570-f001:**
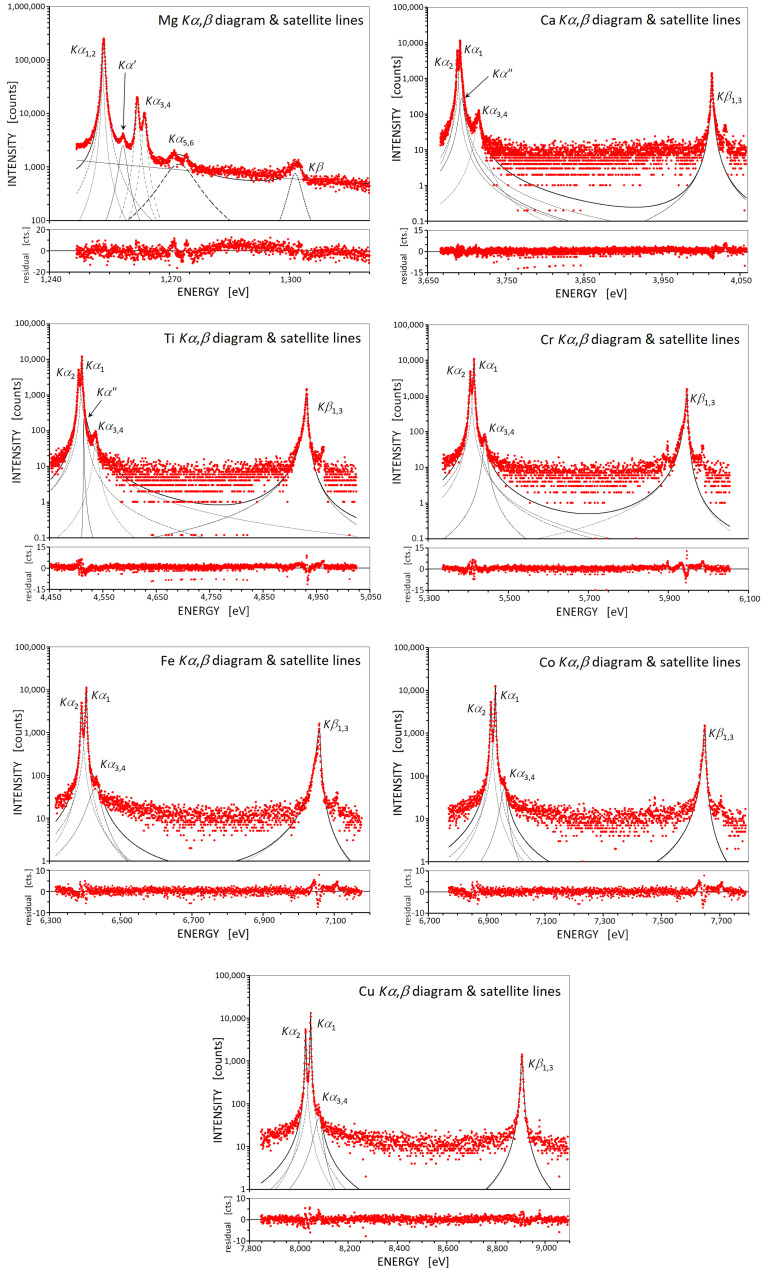
Some (Mg, Ca, Ti, Cr, Fe, Co, and Cu) of the observed Kα,β emission spectra for elements Mg to Cu are shown. The fitting analyses using asymmetric Lorentz functions were performed for elements from Sc to Cu to obtain the Kβ/Kα intensity ratio. For the other elements, symmetric Lorentz function fitting analyses were performed. Visible satellite lines $K\alpha^{\prime \prime }$, $K\alpha_{3,4}$, and $K\alpha_{5,6}$ appear due to shake processes. Note the log scale in the vertical axes.

**Figure 2 ijms-24-05570-f002:**
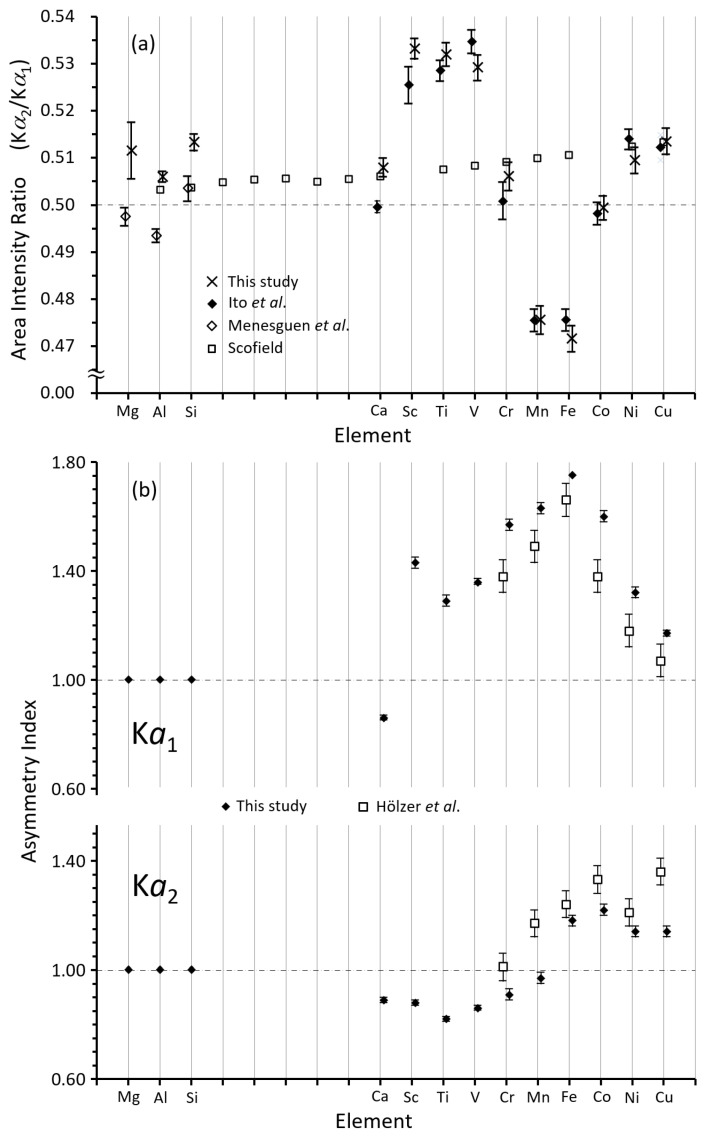
(**a**) $K\alpha_{2}/K\alpha_{1}$ intensity ratio of the elements Mg to Cu. Cross marks are the present experimental data, open square are Scofield’s [3] theoretical data, solid diamonds [6], open diamonds in Mg, Al, and Si [23]. The $K\alpha_{1,2}$ spectral lines in each of the elements were measured using a high-resolution double-crystal X-ray spectrometer [6,20,24]. (**b**) Asymmetry index of $K\alpha_{1,2}$ lines for the elements Mg to Cu are shown together with the data by Hölzer et al. [9]. Asymmetric indices are obtained with an asymmetric Lorentzian fitting analysis [20,21]. See the text for details.

**Figure 3 ijms-24-05570-f003:**
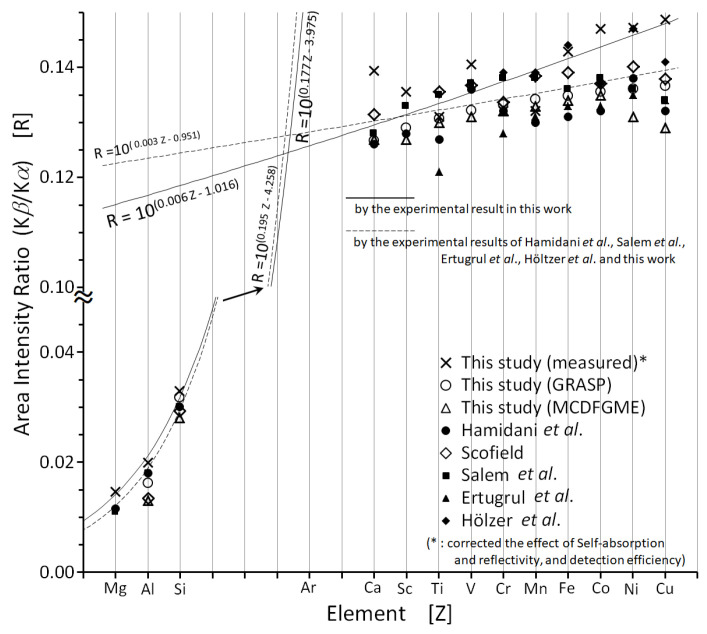
Kβ/Kα intensity ratio from Mg to Cu together with other reported data–Hamidani et al. average [1], Scofield [3], Salem et al. [25], Ertugrul et al. [26], and Hölzer et al. [9]. See text for details.

**Figure 4 ijms-24-05570-f004:**
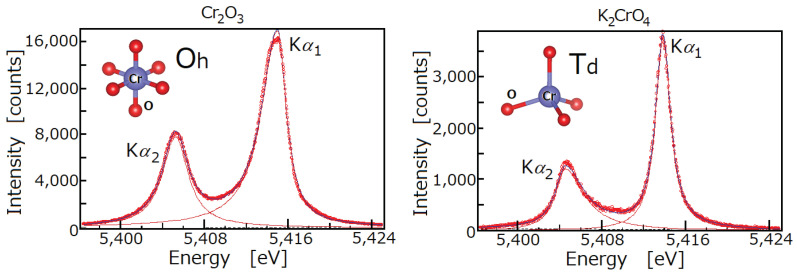
Cr $K\alpha_{1,2}$ spectral lines in Cr${}_{2}$O${}_{3}$ (**left**) and K${}_{2}$CrO${}_{4}$ (**right**) compounds measured using a high resolution double-crystal X-ray spectrometer. Oh symmetry represents ionic bonds, while Td symmetry represents covalent bonds. The spectral profiles show the differences in the energy values, half widths, and asymmetry indices.

**Figure 5 ijms-24-05570-f005:**
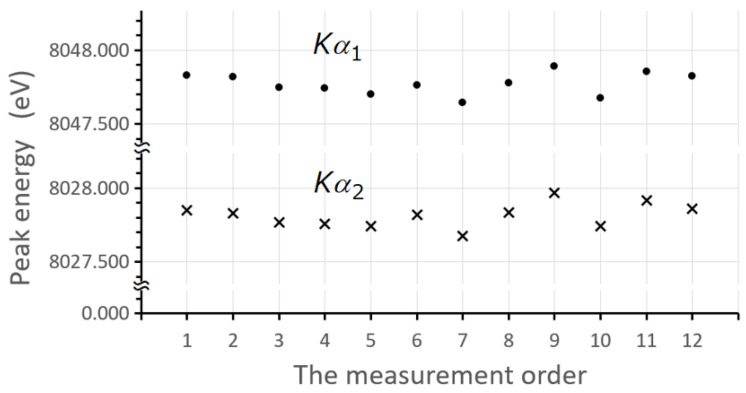
Peak position of the Cu $K\alpha_{1,2}$ lines due to asymmetry fitting in different measurements. For each measurement, a sample of 3d elements was measured. The variation of the peak position is very small. The Kβ/Kα intensity ratio of each element seems to be hardly affected by the variation of the instrument. For details, see the text.

**Table 1 ijms-24-05570-t001:** Kβ/Kα intensity ratio, in %. See text for details.

*Z*	Symb.	Experiment					Theory					
		This Work	Ref. [25]	Ref. [26]	Ref. [1]	Ref. [9]	This Work					Ref. [3]
							GRASP			MCDFGME		
							3*p^n^*	3*d*^*n*−2^4*s*^2^	3*d*^*n*−1^4*s*^1^	No Sat	With Sat	
12	Mg	1.46(2)		1.1	1.15							
13	Al	1.99(4)		1.8	1.8		1.6			1.3	2.2	1.34
14	Si	3.30(3)		3	3.0		3.2			2.8		2.94
20	Ca	13.93(11)		12.8	12.6			12.7		12.7	13.2	13.15
21	Sc	13.53(13)		13.3	12.8			12.9	12.4	12.7		
22	Ti	13.11(5)	12.1	13.5	12.7			13.1	12.6	12.9	13.3	13.55
23	V	14.15(8)		13.7	13.6			13.2	12.8	13.1		13.67
24	Cr	13.20(20)	12.8	13.8	13.2	13.9		13.3	13.0	13.2		13.37
25	Mn	13.24(22)	13.1	13.8	13.0	13.9		13.4	13.1	13.3		13.85
26	Fe	14.37(23)	13.3	13.6	13.1	14.4		13.5	13.2	13.4		13.91
27	Co	14.08(22)	13.3	13.8	13.2	13.7		13.6	13.3	13.5		13.7
28	Ni	14.72(25)	13.5	13.6	13.8	14.7		13.6	13.3	13.1		14.01
29	Cu	14.87(25)	13.4	13.4	13.2	14.1		13.7	13.5	13.2	13.7	13.79

**Table 2 ijms-24-05570-t002:** Line energies and FWHM, in eV, and Asymmetric indexes (AI), resulting from fitting Cu $K\alpha_{1,2}$ emission lines with two asymmetric, and four symmetric Lorentz functions, for the evaluation of the repeat stability of the double-crystal X-ray spectrometer. CF is the corrected FWHM, in eV, by Tochio’s method [24].

2 Asymetric Lorentzian Fitting					
	$K\alpha_{1}$		$K\alpha_{2}$		$\Delta \left(K\alpha_{1}-K\alpha_{2}\right)$
Energy	8047.773±0.075		8027.813±0.081		19.957±0.026
FWHM	2.701±0.007		3.115±0.013		
AI	1.142±0.006		1.158±0.009		
4 symetric Lorentzian fitting					
	$K\alpha_{11}$	$K\alpha_{12}$	$K\alpha_{21}$	$K\alpha_{22}$	$\Delta \left(K\alpha_{11}-K\alpha_{21}\right)$
Energy	8047.773±0.076	8045.207±0.090	8028.038±0.082	8026.476±0.092	19.736±0.024
FWHM	2.422±0.012	2.871±0.084	2.673±0.032	3.260±0.061	
CF	2.273±0.012		2.527±0.032		

**Table 3 ijms-24-05570-t003:** The measurement conditions of Kα,β emission lines in Cr compounds. See text for details.

Sample	Form	Time (s)	
		$K\alpha_{1,2}$	$K\beta_{1,3,5}$
Cr	plate	5	90
Cr${}_{2}$O${}_{3}$	powder	5	150
FeCr${}_{2}$O${}_{4}$	powder	5	150
CoCr${}_{2}$O${}_{4}$	powder	5	150
K${}_{2}$CrO${}_{4}$	powder	6	260
K${}_{2}$Cr${}_{2}$O${}_{7}$	powder	6	150

**Table 4 ijms-24-05570-t004:** Cr $K\alpha_{1,2}$ peak positions, energy difference between the two peaks (eV), half-width (eV), asymmetry index, and Kβ/Kα intensity ratio.

Sample	Symmetry of Cr-O Part	Peak Pos.		Separation	FWHM		Asymm. Index		Int. Ratio
		$K\alpha_{1}$	$K\alpha_{2}$	$K\alpha_{1}-K\alpha_{2}$	$K\alpha_{1}$	$K\alpha_{2}$	$K\alpha_{1}$	$K\alpha_{2}$	Kβ/Kα
Cr metal		5414.7	5405.5	9.2	2.26	2.65	1.65	0.96	0.132
K${}_{2}$CrO${}_{4}$	Td-like	5413.8	5404.6	9.2	1.74	3.30	0.96	0.51	0.156
K${}_{2}$Cr${}_{2}$O${}_{7}$	Td-like	5413.9	5404.6	9.3	1.77	3.31	1.01	0.53	0.159
Cr${}_{2}$O${}_{3}$	Oh-like	5415.0	5405.2	9.8	2.89	2.97	1.77	1.00	0.138
FeCr${}_{2}$O${}_{4}$	Oh-like	5414.9	5405.2	9.7	2.93	2.99	1.81	1.08	0.138
CoCr${}_{2}$O${}_{4}$	Oh-like	5414.9	5405.2	9.7	2.89	3.01	1.79	1.02	0.142

**Table 5 ijms-24-05570-t005:** Comparison of Cr $K\alpha_{1,2}$ peak positions, energy difference between the two peaks (eV), half-width (eV), and asymmetry index of this work and the results of Shuvaev and Kukulyabin [29].

Sample		Peak Pos.		Separation	FWHM		Asymm. Index	
		$K\alpha_{1}$	$K\alpha_{2}$	$K\alpha_{1}-K\alpha_{2}$	$K\alpha_{1}$	$K\alpha_{2}$	$K\alpha_{1}$	$K\alpha_{2}$
Cr metal	This work			9.2	2.26	2.65	1.65	0.96
	Ref. [29]			9.2	1.88	2.43	1.32	1.12
K${}_{2}$CrO${}_{4}$	This work	−0.9	−0.9	9.2	1.74	3.30	0.96	0.51
	Ref. [29]	−0.98	−0.91	9.2	1.65	2.74	0.85	0.64
Cr${}_{2}$O${}_{3}$	This work	0.3	−0.3	9.8	2.89	2.97	1.77	1.00
	Ref. [29]	0.24	−0.16	9.6	2.83	3.04	1.99	1.08

**Table 6 ijms-24-05570-t006:** Experimental conditions in the Kα,β spectral lines measurements. In all cases, the step in 2θ was 0.005°.

Sample	Form	Crystal	Time (s)	Rh (kV, mA)
Mg	plate	ADP(101)	30	40, 60
Al	plate	ADP(101)	30	40, 60
Si	wafer	ADP(101)	50	40, 60
Ca	CaF${}_{2}$ powder	Si(220)	100	40, 60
Sc	foil	Si(220)	6	40, 60
Ti	plate	Si(220)	6	40, 60
V	foil	Si(220)	8	40, 60
Cr	plate	Si(220)	5	40, 60
Mn	plate	Si(220)	5	40, 60
Fe	plate	Si(220)	8	40, 60
Co	foil	Si(220)	5	40, 60
Ni	foil	Si(220)	4	40, 60
Cu	plate	Si(220)	4	40, 60

## Data Availability

Not applicable.

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
