# Peer review of "Intensity Ratio of / in Selected Elements from Mg to Cu, and the Chemical Effects of Cr 1,2 Diagram Lines and Cr / Intensity Ratio in Cr Compounds"

_ijms, 2023, doi:10.3390/ijms24065570_

Round 1

Reviewer 1 Report

Report on paper #ijms-2194648, “Intensity ratio of Kβ/Kα in the elements from Mg to Cu, and the chemical effects of Cr Kα1,2 diagram lines and Cr Kβ/Kα intensity ratio in Cr compounds” byY. Ito et al.

The authors present an interesting work about x-ray emissions lines, combining experiments and calculations. I think the study worth publishing in Int. J. Mol. Sci. as the presentation of their results and their critical discussion with previous ones of the literature are convincing. It is also alwaus good to remind the members of the community that emission bands and intensity ratios can depend of the chemical state of the emitting elements. However, I would suggest some changes, listed below, particularly to better explain some points and give further details.

First, I think that more details should be given in the text on the manner the Ka/Kbeta ratio is obtained. If I understand well the Ka lines is fitted and its intensity, area under the peak, should be one of the fitted parameters. But for the Kbeta emission, it is not possible to apply some fit. Indeed, the Kbeta lines of Mg, Al and Si describe the density of states in a valence band and not in a core level, so this density cannot be represented by a bell shape function; it is also well documented in the literature that the Kbeta line of 3d elements is asymmetric and present low energy secondary peak, which again is against a possible fit. So how the Kbeta intensity is determined?

In the experimental section, it would be good that the authors give the dead time of the counter. Indeed Ar/CH4 detectors have quite long dead times, so some saturation can take place, leading to a loss of count, if the count rate is too high.

The authors correctly point the advantage of working with double-crystal spectrometers, as in this conditions instrumental broadening is minimized. But what about the opening (width and height) of the crystals? If the opening is too large then lines could be slightly deformed leading to disturbed intensity measurements. Is this true for double-crystal spectrometers? Could the authors add a comment regarding this point in the text?

In Figure 1, I think that a log scale of the abscissae would be better to easily see at the same time the Ka and Kbeta emissions, as well as the satellites emissions, which have quite different intensities.

Page 5, “The Kβ/Kα intensity ratio was corrected for self-absorption, crystal reflectivity, and detection efficiency”. This is a bit short and more details should be given in the text. Could the authors give more details? Was the reflectance of the crystals measured or calculated? How? Were the pressure of the counter and the thickness of its window sufficiently well-known, so that the efficiency of the detector is determined with a good precision?

Regarding self-absorption, it is well-known that the emission bands can be distorted by self-absorption in the close to the Fermi level, thus leading to a loss of intensity. In this work, Al, Mg and Si Kbeta are emission bands. So if this effect is not correctly taken into account, then the Kbeta intensity can be underestimated and the Ka/Kbeta ratio not correctly determined.

Table 2. Explain how it is possible to obtain a low uncertainty on the Delta value, starting from larger uncertainty values of Ka energies.

If we consider Ka12-Ka22, the Delta value is 18.7 eV, i.e. 1 eV larger the value indicated in the Table 2. So why is the origin of this difference and why considering (11 - 21) and not (12 - 22)? I have to say that I would have expected the use of the (12 – 21) difference, so explain the (11 - 21) choice.

Caption of Figure 2. Why not writing “Cu sample” instead of “a sample of 3d elements”? As it is written, we could understand that other 3d elements have been considered in this table.

The Kβ/Kα intensity ratio of each element seems to be hardly affected by the variation of the

instrument. I do not understand this sentence of the caption because the figure does not present intensity ratios but only photon energies.

Page 8, “the Kα” satellite, that can be seen, for example, in the Ti Kα, β line of Fig. 1”. With the current presentation of the spectra, such faint line cannot be seen. See my previous remark suggesting the use of a log scale.

Page 11, paragraph between lines 259-272. It seems to me that most of the information given here should rather appear in the experimental section. This will not cut the reading of the discussion section. Moreover, the details “The 2θ step angle of this spectrometer is 0.0005. The measurement time per point for Kβ1,3,5 spectra are 90 s for Cr metal and Cr2O3, 150 s for K2Cr2O7, FeCr2O4, and CoCr2O4, and 260 s for K2CrO4. The 2θ step angle of this spectrometer is 0.002.are given twice: in the text and in the caption of Table 4. Only once would be necessary.

Funding section. “Hubert” with a “t”.

Reviewer 2 Report

Dear Authors!

I have reviewed the article, and I think it can be published with minor revision, since the study was carried out carefully and at a high scientific level. My only remark is the that the title and further in the text, it is mentioned everywhere that elements from Mg to Cu have been studied, however, data for some elements (P, S, Cl, K) are not given. Figure 3 shows data for these elements from other authors (Scoefield), and there are a lot of publications that could be compared with obtained data, (for example: 10.1103/PhysRevLett.57.988; 10.1002/xrs.2712).

Additionally, starting from section 4.2. data are given for chromium compounds, which, in fact, are the subject of a separate study. Why was chromium chosen, why are studies not given for other elements? For manganese and iron, it is easy to find various compounds, in addition, there are also many publications on this topic. Even without this section, the article is of undoubted scientific value.

So, I would you fill the indicated gaps and may be divide the existing material into two parts for two articles. If you find this unacceptable, I nevertheless recommend to indicate why certain elements were excluded from the Mg-Cu list, and why only chromium was chosen for close study.
